# No Cytotoxic and Inflammatory Effects of Empagliflozin and Dapagliflozin on Primary Renal Proximal Tubular Epithelial Cells under Diabetic Conditions In Vitro

**DOI:** 10.3390/ijms21020391

**Published:** 2020-01-08

**Authors:** Patrick C. Baer, Benjamin Koch, Janina Freitag, Ralf Schubert, Helmut Geiger

**Affiliations:** 1Division of Nephrology, Department of Internal Medicine III, University Hospital, Goethe-University, 60596 Frankfurt/M., Germanyjaninafreitag@gmx.net (J.F.); h.geiger@em.uni-frankfurt.de (H.G.); 2Division of Allergology, Pneumology and Cystic Fibrosis, Department for Children and Adolescents, University Hospital, Goethe-University, 60596 Frankfurt/M., Germany; ralf.schubert@kgu.de

**Keywords:** renal tubular cells, epithelial cells, proximal tubule, cytotoxicity, injury, inflammation, empagliflozin, dapagliflozin, kidney

## Abstract

Gliflozins are inhibitors of the renal proximal tubular sodium-glucose co-transporter-2 (SGLT-2), that inhibit reabsorption of urinary glucose and they are able to reduce hyperglycemia in patients with type 2 diabetes. A renoprotective function of gliflozins has been proven in diabetic nephropathy, but harmful side effects on the kidney have also been described. In the current project, primary highly purified human renal proximal tubular epithelial cells (PTCs) have been shown to express functional SGLT-2, and were used as an in vitro model to study possible cellular damage induced by two therapeutically used gliflozins: empagliflozin and dapagliflozin. Cell viability, proliferation, and cytotoxicity assays revealed that neither empagliflozin nor dapagliflozin induce effects in PTCs cultured in a hyperglycemic environment, or in co-medication with ramipril or hydro-chloro-thiazide. Oxidative stress was significantly lowered by dapagliflozin but not by empagliflozin. No effect of either inhibitor could be detected on mRNA and protein expression of the pro-inflammatory cytokine interleukin-6 and the renal injury markers KIM-1 and NGAL. In conclusion, empa- and dapagliflozin in therapeutic concentrations were shown to induce no direct cell injury in cultured primary renal PTCs in hyperglycemic conditions.

## 1. Introduction

Over 90% of the filtered glucose in the kidney is reabsorbed by the sodium-glucose co-transporter 2 (SGLT-2) expressed in the apical brush border membrane of tubular epithelial cells [1,2]. Inhibition of SGLT-2 blocks the reabsorption of glomerularly filtered glucose (and sodium) in the ensuing proximal S1/S2 segment, a mechanism that has been exploited to reduce hyperglycemia in patients with type 2 diabetes mellitus [1]. Based on the selective inhibition of SGLT-2, gliflozins increase urinary glucose excretion by reducing the reabsorption into the bloodstream. Gliflozins are a class of antidiabetic drugs used to lower glucose in type 2 diabetes mellitus [3,4]. The mechanism of action is independent of endogenous insulin and shows a very low risk of hypoglycemia. In the last few years, the renoprotective function of gliflozins has been described in diabetic nephropathy, which affects approximately 40% of patients with diabetes and is a leading cause of chronic kidney disease worldwide. SGLT2 inhibition has been described as reducing inflammation and attenuating the progression of diabetic nephropathy [4].

As the first active substance, the European Medicines Agency granted marketing authorization for dapagliflozin in 2012. Empagliflozin and ertugliflozin have also been approved; in the meantime, canagliflozin has been taken off the market in Germany. Other substances are in advanced clinical trials or are approved in the United States. The pleiotropic effects of SGLT2 inhibitors have the potential to generate benefits beyond the inhibition of glucose reuptake, and there is increasing evidence that gliflozins may reduce the risk of progression of renal impairment in diabetic patients [5]. It has been shown that in patients with type 2 diabetes and kidney disease, the risk of kidney failure and cardiovascular events was lower in the canagliflozin group than in the placebo group [6]. A post-hoc analysis from recent published trial revealed the risk of AKI may be lower under SGLT2 inhibition [7]. In another study of patients with diabetes, initiation of SGLT2 inhibitor therapy has been shown to be associated with a slower rate of kidney function decline and lower risk of major kidney events compared with initiation of other glucose-lowering drugs [8]. Although current evidence supports their safety, additional efforts are needed to elucidate the long-term impact of these compounds on chronic kidney disease, mineral metabolism, and bone health. Indeed, the limited study follow-up precludes a definitive answer on the impact of gliflozins (e.g., on electrolyte and mineral metabolism changes), especially in high-risk subgroups of patients [9]. The findings of ongoing and future clinical trials will help shed further light on the role of gliflozins in the long-term protection of renal function.

Nevertheless, despite overall good clinical tolerability [10], harmful effects also can occur during daily gliflozin intake. Reported adverse events are especially acute kidney injury (AKI) [11,12], acute tubular necrosis [13], genital infections, and bone fractures [3]. Although current evidence supports their safety, additional efforts are needed to elucidate the long-term impact of these compounds on chronic kidney disease, mineral metabolism, and bone health. Indeed, the limited follow-up studies and the heterogeneity of the case-mix of different randomized controlled trials preclude a definitive answer on the impact of these compounds on long-term outcomes such as the risk of bone fracture.

The US Food and Drug Administration (FDA) issued a warning in June 2016 covering the increased risk of AKI following the use of gliflozins [14]. A total of 101 cases were reported between March 2013 and October 2015, but a much higher number is expected worldwide. The FDA warning was based on data suggesting a link between the development of AKI and two approved gliflozins [3]. In this alert, patients with diabetes and heart failure, chronic renal insufficiency, and/or decreased circulating blood volume were included in the risk group. The risk is potentiated by the additional intake of ACE inhibitors, diuretics, and/or nonsteroidal anti-inflammatory drugs, warned the FDA. In November 2019, the German Drug Commission concluded that treatment with gliflozins should be stopped if patients are hospitalized due to major surgery or acute serious illness [15]. However, despite these relevant safety issues, the points are still under investigation and no firm conclusion can be currently drawn.

Possible pathophysiologic mechanisms of AKI induced by SGLT-2 inhibitors have been highlighted recently [16]. Osmotic diuresis can lead to volume depletion, while uricosuria as a consequence of increased glucose–uric acid exchange in the proximal tubule, has the potential of direct tubular damage through crystal formation. High glucose concentrations in renal tubules may lead to altered fructose metabolism in proximal tubular cells and, thus, to local cytotoxicity and tubular injury from increased oxidative stress and the release of inflammatory molecules [16]. Summing up all current data, it can be concluded that a very detailed study of why some patients on gliflozin therapy develop acute renal failure, but not others, and details about the underlying cellular pathomechanisms in the kidney are needed urgently. To date, there is only speculation about the exact pathomechanisms of the renal side effect of gliflozins. No clear investigations have been published to elucidate these mechanisms.

In this project, highly purified and cultured human renal proximal tubular epithelial cells (PTCs) were used as an in vitro model to study the cellular damage of gliflozins. This is the first study using primary highly differentiated PTCs to investigate possible effects of two gliflozins: empa- and dapagliflozin. As PTCs are the direct site of action for SGLT2 inhibitors, a simulation of different environmental variables, such as high glucose, or combinations with other drugs were used to investigate the influence on the renal tubule system in the in vitro model and possibly to simulate the conditions in the human nephron.

## 2. Results

### 2.1. Characterization of SGLT-2 Expression and Function in PTCs

Immunofluorescence staining clearly showed the expression of SGLT-2 in the apical membrane of cells of the proximal tubule in situ (Figure 1A) and PTCS in vitro (Figure 1C). The staining of human renal sections showed that SGLT-2 is exclusively expressed in the apical brush border membrane of tubular epithelial cells.

Cultured PTCs displayed an epithelial morphology with a highly compact cell monolayer (Figure 1B). The expression of SGLT2 mRNA and protein in cultured PTCs was also proven by standard PCR analysis (Figure 1D) and western blotting (Figure 1E). The RNA and protein isolated from human kidneys (renal extracts) were used as positive controls in both verifications (Figure 1D,E).

As a functional assay, the glucose uptake was evaluated using the fluorescent glucose analog NBDG-2. Loading of NBDG-2 into PTCs was shown by the measurement of unloaded cells as a negative control. The SGLT2-mediated NBDG-2 influx from the cell culture medium into the cells was sensitive to inhibition by the SGLT inhibitors in a dose-dependent manner (Figure 2). Incubation of PTC with 10 nM empa- or dapagliflozin had no inhibitory effect on the glucose uptake of PTCs. The NBDG-2 assay confirms the activity of the sodium-glucose co-transporter in cultured PTCs and the inhibitory effect of the gliflozins.

### 2.2. Cell Viability, Proliferation, and Cytotoxicity Assays

We used two viability assays using calcein-acetoxymthyl (calcein-AM) and XTT to investigate the effects of empa- and dapagliflozin on PTCs viability. Calcein-AM is a cell-permeant non-fluorescent dye, which is exclusively in viable cells converted into calcein, a dye with intense green fluorescence. The XTT assay is a colorimetric assay used to estimate the metabolic activity of viable cells.

Both assays demonstrated that empa- and dapagliflozin induce no cytotoxic effects on cultured PTCs. Depending on the experiment, media were additionally supplemented by a combination of empa- or dapagliflozin with ramipril or hydro-chloro-thiazide (HCT). Nevertheless, no statistically significant difference could be detected compared to the control (high glucose medium: HG) either with the calcein assay (Figure 3A) or the XTT assay (Figure 3B). In addition, we used different internal negative controls within the two assays. In the calcein assay, we used a mixture of cytokines which significantly reduced the viability of PTCs over the incubation period (43.5 ± 8.4% versus HG (=100%)) (Figure 3A). For the XTT, we used a low glucose medium 199 (LG) as a control. Incubation in LG resulted in a reduced metabolic activity, due to reduced cell proliferation (72.2 ± 5.5% versus HG (=100%)) (Figure 3B).

In addition, both a proliferation and a cytotoxicity assay were used to detect further effects of empa- and dapagliflozin on PTCs. The fluorometric assay with 4,6-diamino-2-phenylindole (DAPI), measuring the DNA content as an indirect determination of cell number and proliferation, verified that both gliflozins had no influence on cell proliferation. Furthermore, the combinations with ramipril or HCT also showed no effects on cell proliferation, whereas the negative control (LG) reduced proliferation of PTCs significantly (77.8 ± 2.5% versus HG (=100%)) (Figure 4A). Cytotoxic effects, measured by quantification of lactate dehydrogenase (LDH) in the supernatant of PTCs after incubation in gliflozin-containing media or co-medications, were also not detectable (Figure 4B). Only the incubation with the cytomix induced a significant increase in LDH activity in the supernatant (148.3 ± 16.7% versus HG (=100%)) (Figure 4B).

### 2.3. Measurement of Oxidative Stress and Tubular Injury Markers

Formation of intracellular oxidative stress was proven using fluorescence measurements of intracellular 2′,7′-dichlorofluorescein (DCF). Incubation of PTCs in HG increased oxidative stress levels in a significant manner. Addition of empagliflozin (500 nM) to HG could not reduce the oxidative stress level (Figure 5). By contrast, oxidative stress generation evoked by high glucose exposure was significantly suppressed by the treatment with dapagliflozin (500 nM). We further checked the effect of H_2_O_2_ (250 µM) after loading for establishment of the DCF measurements, resulting in a maximum signal (positive control, data not shown). Unloaded cells were used as negative controls and compared with cells after DCF loading (data not shown).

We further studied the role of SGLT2 inhibition in the induction and release of pro-inflammatory and injury markers. Firstly, we checked the mRNA levels of the pro-inflammatory cytokine interleukin-6 (IL-6) and two renal tubular injury markers (kidney injury molecule-1 (KIM-1), and neutrophil gelatinase-associated lipocalin (NGAL)) after 24 h incubation in HG with gliflozins. As shown in Figure 6, neither empagliflozin nor dapagliflozin influenced the mRNA expression of all three readouts. Comparison between LG and HG showed an induction of IL-6 and NGAL mRNA after incubation for 24 h, but not of KIM-1 (data not shown). Stimulation with a mixture of cytokines was used as a positive control. The cytomix induced a significant induction of IL-6 and NGAL mRNA but not KIM-1 mRNA expression (Figure 6).

We then checked the release of all three molecules in the supernatant of PTCs (Figure 7). At the protein level, IL-6 is constitutively released by PTCs cultured in HG (11.9 ± 1.7 ng/mL; mean ± SD, *n* = 6). When the cells were incubated with gliflozins, no significant effect on the release of IL-6 protein was detected. Incubation in the presence of the cytomix increased the release of IL-6 protein significantly (417 ± 117 %, calculated as percent in relation to the control (HG), *n* = 6). In addition, we also found no effect of empa- or dapagliflozin on the release of KIM-1 and NGAL. The cytomix induced the release of NGAL protein in a significant manner (269 ± 57 %, *n* = 4), whereas release of KIM-1 was not increased. At the protein level, KIM-1 and NGAL are also constitutively released by PTCs cultured in HG (11.6 ± 2.1 ng/mL and 115.7 ± 26.2 ng/mL, respectively; mean ± SD, *n* = 4).

## 3. Discussion

Gliflozins are the latest class of hypoglycemic agents for the treatment of diabetes mellitus type 2 and have a unique mechanism of lowering glucose by the inhibition of renal proximal tubular sodium-glucose co-transporter-2. SGLT-2 inhibitors have been shown to increase urinary glucose excretion and decrease serum glucose and HbA1c. In addition, compared to other topical antidiabetic medications, gliflozin intake can also reduce body weight. The drug group is further attributed to an improvement in hypertension, a reduction in cardiovascular mortality risk, and multiple effects on the kidney. Nevertheless, despite good overall clinical tolerability, several harmful effects have been published related to daily intake. Caution is still advised regarding possible adverse events described such as AKI and acute tubular necrosis [3,13]. Under certain circumstances—possibly dehydration of the patient, low blood pressure, or certain co-medications—gliflozin may damage the organ. Currently, however, there is only speculation about the exact pathomechanisms of the renal side effects [16]. No clear investigations have been published to elucidate these mechanisms. The influence of osmotic pressure and the development of excess uric acid on the function of tubular epithelial cells in the nephron might be involved in this context [18]. Increased osmotic pressure (hyperosmolarity) in the tubular system may activate the polyol pathway in epithelial cells [19], leading to osmotic damage via the accumulation of sorbitol and the formation of oxidative stress with cellular damage, inflammation, and pathological tissue change [20,21]. The main site of action of SGLT-2 inhibition raises the question whether inhibition of the transporter induces direct cytotoxic or inflammatory effects on renal proximal tubular epithelial cells.

In the current study, we focus on the effects of two currently available gliflozins on renal proximal tubular epithelial cells. For the first time, we examined the cytotoxic and inflammatory effects of empa- and dapagliflozin on primary and highly differentiated PTCs under diabetic conditions in vitro. High levels of circulating glucose and inflammation are the main causes of tissue damage in diabetes mellitus by inducing cell injury and high levels of oxidative stress in the kidney [22]. In this case, it should also be mentioned that diabetes itself is a systemic pro-inflammatory condition, involving not only the kidney, but also the cardiovascular system and bone [23,24].

Several in vitro studies using immortalized human tubular epithelial cells (e.g., HK2 cell line) [25,26,27,28], murine tubular epithelial cells [29] or non-renal cells [30,31] investigated either the effects of empa- or of dapagliflozin. Some of them have shown that SGLT2 inhibition reduced the release of inflammatory or fibrotic factors induced by high glucose levels [25,26,27], others found effects on oxidative stress responses [26,30,31]. Our current study looked at whether high glucose induced changes were influenced by either of the two gliflozins. Whereas we found no effect on cell viability, proliferation, and cytotoxicity, a significant reduction in the oxidative stress was found after incubation with dapagliflozin. Chen and co-workers also demonstrated decreased reactive oxidative species after incubation with dapagliflozin in a fructose-induced diabetic milieu [31]. Others investigated the subcellular mechanisms underlying the protective effects of empagliflozin from high glucose-mediated injuries on HK-2 cells [26]. As oxidative stress plays pathologic roles in diabetic kidneys [32] and empagliflozin may reduce this effect [33], the authors analyzed the effects of empagliflozin and described the protection of HK-2 cells from high glucose-mediated injuries through a mitochondrial mechanism.

In addition, a series of in vitro cytotoxicological investigations were conducted to evaluate a mode of action for gliflozin-associated effects. In addition to the in vitro studies, the effects of gliflozins on the kidney were also assessed in several in vivo studies and in clinical trials (currently reviewed by Ninčević and co-workers [34]), which should not be discussed here. Cytotoxicity and inflammation mediated by high glucose is directly responsible for pathological changes in diabetic nephropathy [35]. The in vitro data presented by Smith and co-workers demonstrated no genotoxic, cytotoxic, or mitogenic role for empagliflozin. However, the data indicate that an oxidative metabolite of empagliflozin is cytotoxic to renal tubular cells, but is not genotoxic [29]. Others described a canagliflozin-induced cytotoxicity at particularly low concentrations in proliferating immortalized tubular epithelial cells in vitro [36]. We also checked the influence of often used co-medications, such as ramipril or HCT, on the cytotoxicity and viability of PTCs. Co-incubation of gliflozins with either ramipril, clinically used to lower hypertension via inhibition of the renal angiotensin-converting enzyme, or HCT, clinically used as diuretic medication, did not induce significant effects on PTCs. We did not check the co-medications by PCR and enzyme-linked immunosorbent assay (EIA) due the negative results of the cytotoxicity and viability assays (and also to the low availability of primary isolated PTCs). Measurement of the inflammatory marker (IL-6) and two injury markers (KIM-1 and NGAL) showed that the gliflozins did not influence their expression in vitro. In contrast to our study, other have shown that SGLT-2 inhibition with empagliflozin reduces high glucose-induced IL-6 release in HK-2 cells [27]. Dekkers and co-workers further described in a post-hoc analysis of a clinical trial that dapagliflozin decreased urinary KIM-1 and IL-6 excretion, suggesting that dapagliflozin may beneficially affect renal inflammation and reduces tubular cell injury [37].

In summary, our current study describes the expression and function of SGLT-2 on primary, highly differentiated PTCs and investigated the effects of SGLT-2 inhibition using two gliflozins: empa- and dapagliflozin. Cell viability, proliferation and cytotoxicity assays revealed that neither empagliflozin nor dapagliflozin induces effects in PTCs cultured in diabetic medium, or co-medication with ramipril or HCT. Oxidative stress was significantly lowered by dapagliflozin but not empagliflozin. No effect of either inhibitor could be detected on the mRNA and protein expression of inflammatory or injury markers. In conclusion, empa- and dapagliflozin in therapeutic concentrations (C_max_ 500 nM) induce no direct cell injury in cultured primary renal proximal tubular cells. Nevertheless, further future studies are needed to investigate potential specific factors leading to tubular injury. First, the effects of incubation in high concentrations of fructose and in hyperosmotic media, both in the presence of gliflozins and high glucose media, should be investigated. High fructose, in vivo generated due to consumption of high fructose dietary products, may also affect tubular epithelial cells [38]. Dietary fructose has been shown to cause tubular injury in vitro and in vivo [21,39]. In addition, an increment in osmotic pressure, generated in vivo by dehydration, may cause activation of the polyol pathway also leading in the accumulation of fructose [16,20]. On the other hand, high concentrations of uric acid could be another factor involved [18], whereas it has been shown that canagliflozin therapy decreased serum uric acid in patients with type 2 diabetes [40]. Finally, the in vitro effects of supratherapeutic gliflozin concentrations that may mimic the accumulation of the drug in conditions of renal failure in vivo should also be investigated in future studies.

## 4. Materials and Methods

### 4.1. Isolation and Culture of Primary Human Renal Proximal Tubular Epithelial Cells

Human renal tissue was obtained from patients undergoing tumor nephrectomies. The donors gave written informed consent. The study was approved by the ethics committee of the clinic of the Goethe University, Frankfurt (05 Dec 2014, UGO 03/10, Amendment). The ethical standards defined by the World Medical Association Declaration of Helsinki were complied with.

Primary human renal proximal tubular epithelial cells (PTC) were separated as described previously [41]. In brief, cells were isolated after tumor nephrectomies from renal tissue not involved in renal cell carcinoma. The tissue was disintegrated using crossed blades, digested with collagenase/dispase, and passed through a 106 µm mesh. Remaining cohered cells were then incubated with collagenase IV, DNase, and MgCl_2_ and further purified by a Percoll density gradient centrifugation. The cells were then isolated by a mAb against aminopeptidase M (CD13) and the Mini-MACS system (Miltenyi, Bergisch Gladbach, Germany) to enrich highly purified PTC. Primary isolated PTC were strongly positive for aminopeptidase M (98.6 %). Ultrastructural analysis revealed highly preserved brush border microvilli, a well-developed endocytosis apparatus, and numerous mitochondria [41,42]. Isolated cells were seeded in 6-well plates. Medium 199 (M4530, Sigma-Aldrich, Taufkirchen, Germany) with a physiologic glucose concentration (100 mg/dl) was supplemented with 10% fetal bovine serum (FBS; Biochrom, Berlin, Germany), used as standard culture medium and replaced every three to four days. Confluent cells were passaged by trypsinization. Cells between passages 2 and 5 were used for the experiments.

### 4.2. Characterization of SGLT-2 Expression

Regarding histological staining, a small portion of human renal tissue normally used for PTC isolation was fixed with 10% formalin at room temperature for 4–6 h. Subsequently, paraffin-embedded tissue sections (4 µm) were dewaxed and rehydrated in xylol, 100, 96, and 70 % ethanol, boiled in TRIS-buffer (10 mM Tris Base, 1 mM EDTA, 0.05 % Tween 20, pH 9.0) for 20 min for antigen retrieval and washed with phosphate-buffered saline (PBS). Samples were blocked with normal goat serum at 37 °C for 30 min and incubated with anti-SGLT-2 (Santa Cruz Biotechnology, Heidelberg, Germany, no. sc-393350, 1:50) overnight at 4 °C. After washing with PBS, a Cy3-conjugated goat-anti-mouse IgG (Jackson Immuno Research, Cambridgeshire, UK, no. 115-165-062; 1:300) was applied for 1 h at 37 °C. Sections were subsequently, mounted in mounting medium and examined using Zeiss fluorescence microscope equipment.

Regarding fluorescence microscopy, PTC cultured on chamber slides was rinsed three times with PBS, and fixed with paraformaldehyde (4%) for 10 min. The fixed cells were washed twice. Unspecific binding sites were blocked by PBS containing 5% normal goat serum for 20 min. Primary antibody anti-SGLT-2 (Santa Cruz Biotechnology, no. sc-393350; 1:50) was applied without washing and incubated for 30 min at 37 °C with gentle shaking. After washing with PBS, cells were incubated with a Cy3-conjugated goat-anti-mouse IgG (1:300) for 30 min at 37 °C. Controls of autofluorescence or non-specific fluorescence were performed on fixed cells processed without the primary antibody. Monolayers were mounted in mounting medium and examined using Zeiss fluorescence microscope equipment.

Proof of SGLT-2 mRNA expression in cultured PTCs was done by PCR analysis (described in Section 4.7). The expression of SGLT-2 protein on PTC was examined using western blotting, as described previously [43]. In brief, the cells were lysed using 10 mM Tris pH 7.4, 0.1% sodium dodecyl sulphate (SDS), 0.1% Tween20, 0.5 % TritonX100, 150 mM NaCl, 10 mM EDTA, 1 M urea, 10 mM NEM, 4 mM benzamidine, and 1 mM PMSF and collected by scraping. After centrifugation, the pellet was suspended in Laemmli’s buffer and heated at 95 °C for 5 min prior to electrophoresis on a 10 % SDS polyacrylamide gel. The protein content was determined by a standard assay and an equal volume of protein was loaded into each lane. The separated proteins were transferred electrophoretically to Immobilon transfer membrane (Millipore). Membranes were blocked for 2 h. Immunoblotting was performed by incubating with antibodies against SGLT-2 (resulting in a 73 kDa band; Santa Cruz Biotechnology, no. sc-393350; 1:200), followed by a secondary antibody (horseradish peroxidase-conjugated anti-mouse IgG; DAKO P0447, 1:1000). Protein bands were made visible using the Peqlab Fusion FX system (VWR, Darmstadt, Germany).

### 4.3. Stimulations

Cells were grown in 24- or 96-well culture plates, or small culture flasks (25cm^2^) in standard cell culture medium (described in Section 4.1) until confluence. PTCs in selected experiments were grown to subconfluence and used for the assays (as indicated). The cells were then washed and kept in serum-free medium 199 for 2 h and stimulated as indicated in the particular assay. In most assays, LG with 10 % FBS (glucose content 100 mg/dl (5.5 mM)) was used as a negative control. Stimulations were done in HG with 10% FBS (glucose content 450 mg/dl (25 mM)) which induced a diabetic milieu in the cultures. In case of protein measurements in the supernatants (EIA, Section 4.8), stimulations were done in high glucose medium 199 without 10% FBS. For stimulations, empagliflozin (Adipogen, San Diego, CA, USA; no. AG-CR1-3619) was dissolved in dimethylsulfoxide and diluted in medium (500, 100, 10 nM). Dapaglifozin (Cayman Chemical, Ann Arbor, MI, USA; no. 11574) was dissolved in dimethyl sulfoxid and diluted in medium (500, 100, 10 nM). Maximal concentration of both gliflozins (500 nM) is matched to the published maximal therapeutic concentration observed (C_max_) [44,45]. The interactions of gliflozin with ramipril (Cayman Chemicals, no. 15558; stock solution 1mM in dimethyl sulfoxide) or HCT (Sigma-Aldrich, Taufkirchen, Germany, no. H-4759; stock solution 100mM in dH_2_O) were investigated in selected experiments (viability, cytotoxicity, and proliferation assays).

### 4.4. Functional Assay

We used 2-Deoxy-2-[(7-nitro-2,1,3-benzoxadiazol-7-yl)amino]-D-glucose (NBDG-2; Hoelzel Diagnostika, Köln, Germany, no. M6327), a fluorescent glucose analogon for usage in fluorescence measurements [46], to monitor glucose uptake in cultured PTCs. Confluent PTCs in 96-well plates were washed with pre-warmed HBSS. The HBSS containing NBDG-2 (100 µg/mL) either alone or with decreasing concentrations of gliflozins (500, 100, 10 nM) was added for 60 min at 37 °C (in quadruplicate for each biological replicate). Cells incubated in buffer without NBDG-2 were used as background controls. The cells were then washed with HBSS and fluorescence was measured directly using a fluorescence reader (BMG Fluostar, Ortenberg, Germany) with excitation and emission wavelengths of 485 nm and 538 nm, respectively. Data are expressed as arbitrary fluorescence units.

### 4.5. Cell Viability, Proliferation, and Cytotoxicity Assays

The cell viability of PTC was determined by two viability assays, a photometric assay using 2,3-Bis-(2-Methoxy-4-Nitro-5-Sulfophenyl)-2*H*-Tetrazolium-5-Carboxanilide (XTT) and a fluorescence-based assay using calcein-AM (Biolegend, San Diego, CA, USA) to investigate any possible cytotoxic effects, as described previously [43]. In brief, confluent PTCs in 96-well plates were incubated for 6 d in media containing empa- or dapagliflozin and co-medications (in quintuplicate for each biological replicate). The XTT reagent was then added to the wells, as described by the manufacturer (AppliChem, Darmstadt, Germany), and incubated at 37 °C for 4 h. Absorbance was measured in an Apollo LB911 microplate reader (Berthold, Bad Wildbad, Germany) at 492 nm vs. 650 nm. For the fluorescence assay, cells were washed and calcein-AM (1 µM) was added incubated at 37 °C for 30 min. Fluorescence was then measured immediately using a fluorescence reader (BMG Fluostar, Ortenberg, Germany) with excitation and emission wavelengths of 485 nm and 515 nm, respectively. Cells incubated in buffer without calcein-AM were used as background controls. Data are calculated as a percent in relation to the control (HG). Cells cultured in LG were used as a negative control.

Epithelial cell proliferation was determined by a fluorometric assay using DAPI, measuring the DNA content as an indirect determination of cell number and proliferation [17]. In brief, subconfluent PTCs in 96-well plates were incubated for 6 d in media containing empa- or dapagliflozin and co-medications (in quintuplicate for each biological replicate). Then, cells were permeabilized using 0.02% SDS, 150 mM NaCl, and 15 mM sodium citrate. Finally, DAPI (2 μg/mL) was added to each well. Fluorescence was measured in a fluorescence reader (FluoStar, BMG Labtech, Offenburg, Germany) with excitation and emission wavelengths of 355 nm and 460 nm, respectively. Data are calculated as a percent in relation to the control (HG). Cells cultured in LG were used as a negative control.

The measurement of LDH activity released from the cytosol of damaged cells into the supernatant was used for the quantification of cell death and cell lysis to determine cytotoxic effects. Supernatants were harvested after incubation in gliflozin-containing media for 6 d and processed as described by the manufacturer (Sigma-Aldrich, Taufkirchen, Germany, no. 11644793001). Absorbance was measured in a microplate reader at 490 nm vs. 650 nm. Data are calculated as a percent in relation to the control (HG). Cells cultured in medium 199 containing a mixture of cytokines (Cytomix: γ-interferon, 200 U/mL; IL-1β, 25 U/mL; and tumor necrosis factor-α, 10 ng/mL) were used as a positive control.

### 4.6. Measurement of Oxidative Stress

The cell-permeant 2′,7′-dichlorodihydrofluorescein diacetate (H_2_DCFDA) was used to detect reactive oxygen species in PTCs. Upon cleavage of the acetate groups by intracellular esterases and oxidation, nonfluorescent H_2_DCFDA is converted to highly fluorescent DCF. Confluent PTCs in 96-well plates were cultured in LG, HG or with gliflozins (500 nM) for 6 d. The cells were then washed with pre-warmed HBSS and H_2_DCFDA (20 µM in HBSS) was added for 30 min at 37 °C (in quintuplicate for each biological replicate). Cells incubated in buffer without DCF were used as background controls. The cells were then washed with HBSS and fluorescence was measured immediately using a fluorescence reader (BMG Fluostar, Ortenberg, Germany) with excitation and emission wavelengths of 485 nm and 538 nm, respectively. Data are expressed as arbitrary fluorescence units.

### 4.7. PCR

The RNA extraction was performed using single-step RNA isolation from cultured PTC by a standard protocol. After the RNA extraction, cDNAs were synthesized for 30 min at 37 °C using 1 µg RNA, 50 µM random hexamers, 1 mM deoxynucleotide-triphosphate-mix, 50 units of reverse transcriptase (Fermentas, St. Leon-Rot, Germany) in 10× PCR buffer, 1 mM β-mercaptoethanol and 5 mM MgCl_2_. A Hot FIREPol EvaGreen Mix Plus was used (Solis Biodyne, Tartu, Estonia) for the master mix; the primer mix and RNAse-free water were added. Quantitative PCR was carried out in 96-well plates using the following conditions: 12 min at 95 °C for enzyme activation, 15 s at 95 °C for denaturation, 20 s at 63 °C for annealing, and 30 s at 72 °C for elongation (40 cycles). Finally, a melting curve analysis was conducted. Products were checked by agarose gel electrophoresis in selected experiments. The PCR fragment quantification was realized using the ABI Prism ^®^ 7900HT Fast Real-Time PCR System with a Sequence Detection System SDS 2.4.1 (Thermo Fisher Scientific, Waltham, MA, USA). Relative quantification was estimated by the ∆∆CT method [47] with β-actin as a housekeeper. The level of target gene expression was calculated using 2^−∆∆*C*t^. The PCR products in selected experiments were separated by agarose electrophoresis (2 %) and observed under ultraviolet illumination. Primer pairs were synthesized by Invitrogen (Karlsruhe, Germany) and are listed in Table 1.

### 4.8. Immunoassays

After stimulation for 48 h (described in Section 4.3), supernatants were harvested, centrifuged at 300× *g* for 5 min and used for the quantification of IL-6, KIM-1 or NGAL, or stored at −20 °C for later measurement. Interleukin-6 was quantified using a commercially available EIA kit (Immunotools, Friesoythe, Germany; no. 31670069). KIM-1 was quantified using a commercially available EIA DuoSet (R&DSystems, Wiesbaden, Germany; no. DY1750). NGAL was quantified using a commercially available EIA DuoSet (R&DSystems, Wiesbaden, Germany; no. DY1757). All assays were processed as described by the manufacturer. In brief, the wells of 96-well microtiter plates were coated with the capture antibody overnight at room temperature. Nonspecific binding sites were blocked with blocking buffer for 1 h. The plates were then washed with PBS/0.05 % Tween and the standard (IL-6 assay: 8–500 pg/mL; KIM-1: 15.5–1000 pg/mL; NGAL: 78.1–5000 pg/mL), and the samples were added for 2 h at room temperature. All samples were diluted in assay buffer (IL-6: 1:25; KIM-1: 1:20; NGAL 1:20) and run in duplicate. The plates were washed and incubated with biotinylated detection antibody for 2 h at room temperature, washed again and incubated with horseradish-peroxidase-streptavidin for 30 min. After washing, TMB was added for 5–20 min and the substrate reaction was stopped and measured (450 vs. 620 nm). The data are calculated as ng/mL in the supernatant and presented as percent versus the related control.

### 4.9. Statistical Analysis

The data are expressed as mean ± SD. Analysis of variance with Dunnett’s Multiple Comparison Test or Student’s *t*-test were used for statistical analysis. Analyses were performed using Prism 5.0 (GraphPad Software, San Diego, CA, USA). *p* values < 0.05 were considered significant.

## Figures and Tables

**Figure 1 ijms-21-00391-f001:**
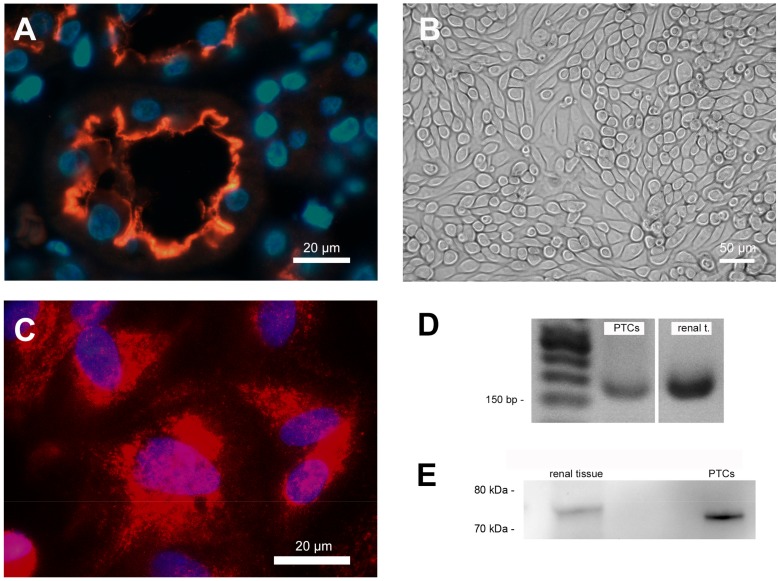
Cell characterization. (**A**) Immunofluorescence staining of human renal paraffin-embedded tissue sections showing sodium-glucose co-transporter 2 (SGLT-2) expression on the apical cell membrane in the proximal tubular nephron. Nuclei were counterstained with 4,6-diamino-2-phenylindole (DAPI; blue). (**B**) Characteristic phase contrast microscopy of confluent PTCs cultured in standard cell culture. (**C**) Immunofluorescence staining of SGLT-2 expression in cultured PTCs. Nuclei were counterstained with DAPI (blue). (**D**) Proof of SGLT-2 mRNA expression in cultured PTCs and RNA from human renal tissue extracts using PCR. (**E**) Proof of SGLT-2 protein expression in cultured PTCs and protein from human renal tissue extracts using western blotting.

**Figure 2 ijms-21-00391-f002:**
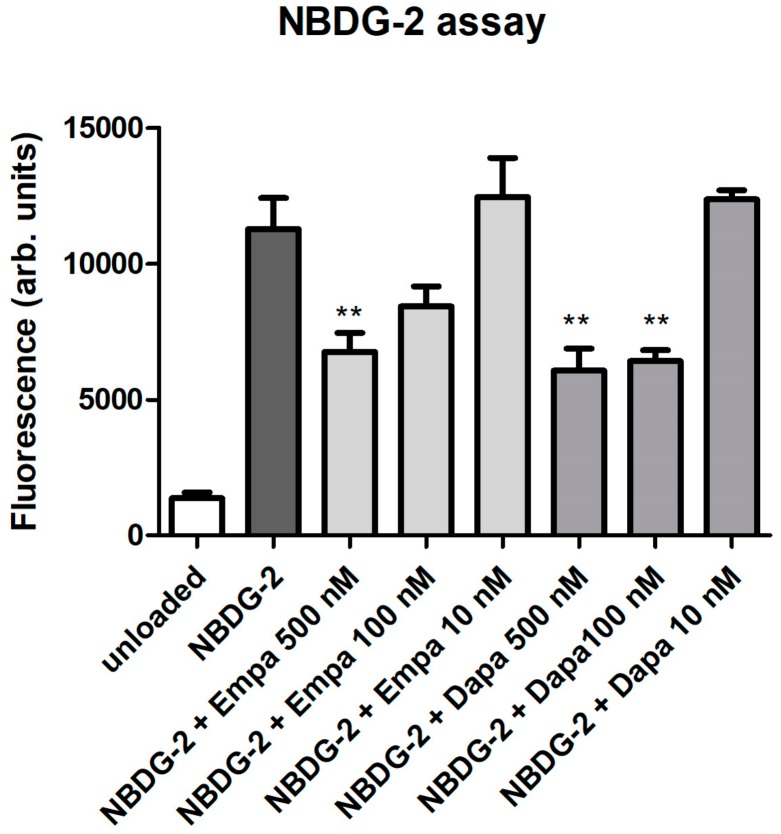
Functional assay. NBDG-2, a fluorescent glucose analog, was used to monitor glucose uptake in cultured PTCs. NBDG-2 was added either alone or with decreasing concentrations of gliflozins (500, 100, 10 nM) for 60 min at 37 °C. Cells incubated in buffer without NBDG-2 were used as background controls (unloaded). Fluorescence was measured using a fluorescence reader with excitation and emission wavelengths of 485 and 538 nm (arbitrary units, mean ± standard deviation (SD), *n* = 4, ** *p*<0.01 versus NBDG-2).

**Figure 3 ijms-21-00391-f003:**
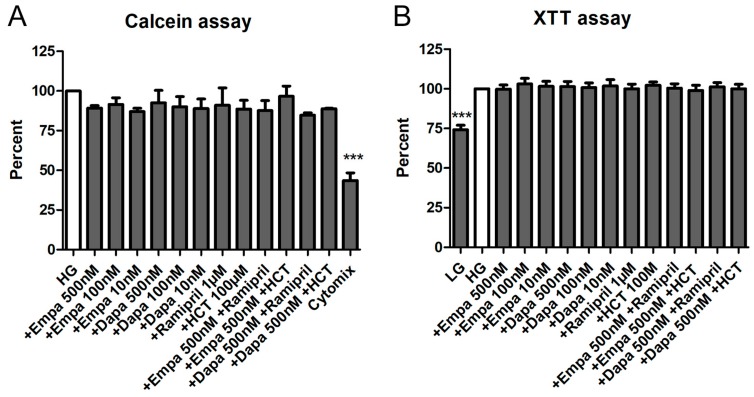
Cell viability assays. The PTCs cultured in 96-well plates were incubated for 6 d in media containing empa- or dapagliflozin and co-medications. No significant effects of the different pretreatments on the cell viability were detected. (**A**) Calcein assay. After loading with calcein-AM for 1h, fluorescence was measured immediately using a fluorescence reader (ex 485 nm, em 538 nm, arbitrary units, calculated as percent in relation to the control (high glucose medium: HG), mean ± SD, *n* = 5). Cells incubated with a mixture of cytokines (Cytomix) were used as a negative control (*** *p* < 0.001). (**B**) XTT assay. The XTT assay was performed and optical density (OD) was measured in a microplate reader at 490 vs. 650 nm (arbitrary units, calculated as percent in relation to the control (HG), mean ± SD, *n* = 4). Cells cultured in low glucose medium 199 (LG) were used as a negative control (*** *p* < 0.001).

**Figure 4 ijms-21-00391-f004:**
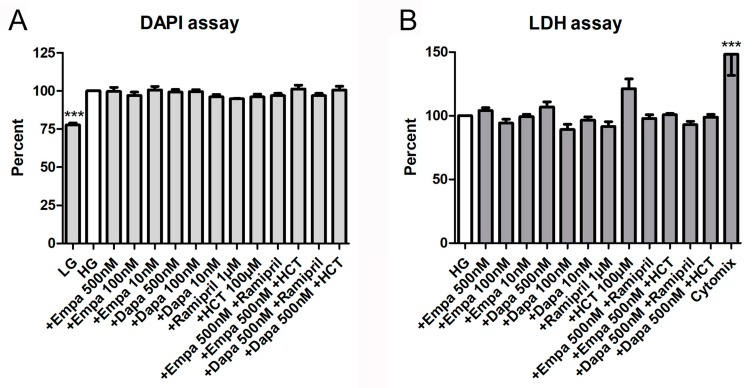
Cell proliferation and cytotoxicity assays. (**A**) DAPI assay. The assay was used to measure the DNA content as an indirect determination of cell number and proliferation [17]. Fluorescence was measured using a fluorescence reader (ex 355 nm, em 460 nm, arbitrary units, calculated as percent in relation to the control (HG), mean ± SD, *n* = 4). Cells cultured in LG were used as a negative control. (**B**) Lactate dehydrogenase (LDH) assay. The measurement of LDH activity released from the cytosol of damaged cells into the supernatant was used for the quantification of cell death and cell lysis. Absorbance was measured in a microplate reader (490 vs. 650 nm, arbitrary units, calculated as percent in relation to the control (HG), mean ± SD, *n* = 4–6). A mixture of pro-inflammatory cytokines was used as a positive control (*** *p* < 0.001).

**Figure 5 ijms-21-00391-f005:**
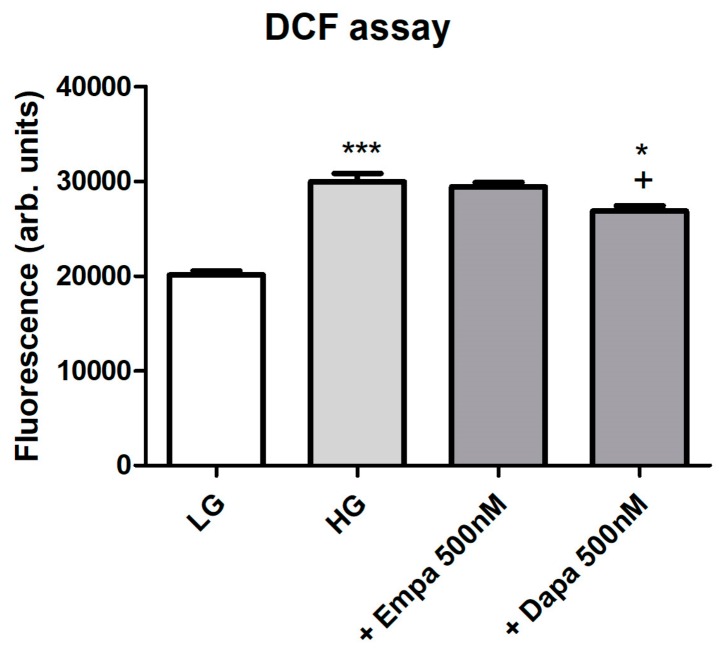
Formation of oxidative stress under diabetic culture conditions. Confluent PTCs in 96-well plates were cultured in LG, HG or with gliflozins (500 nM) for 6 d. Then, 2′,7′-dichlorodihydrofluorescein diacetate (20 µM in Hank’s buffered saline solution) was added for 30 min at 37 °C. Fluorescence of intracellular DCF was measured using a fluorescence reader with excitation and emission wavelengths of 485 and 538 nm (arbitrary units, mean ± SD, *n* = 6, *** *p* < 0.001 vs. LG, + *p* < 0.05 vs. HG, * *p* < 0.05 vs. Empa 500 nM).

**Figure 6 ijms-21-00391-f006:**
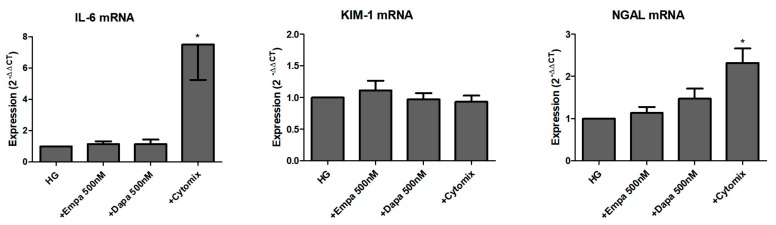
Effect of empagliflozin and dapagliflozin on mRNA expression of selected factors. Cells were grown in small culture flasks until confluence, serum depleted for 2 h and stimulated in HG for 24 h. The expression levels in each experiment were normalized to a housekeeping gene (β-actin) and are expressed relative to the control using the ∆∆CT method. A mixture of pro-inflammatory cytokines (cytomix) was used as a control, whereas only IL-6 mRNA was significantly increased after incubation with the cytomix (* *p* < 0.05 versus HG; IL-6 and NGAL *n* = 6; KIM-1 *n* = 7).

**Figure 7 ijms-21-00391-f007:**
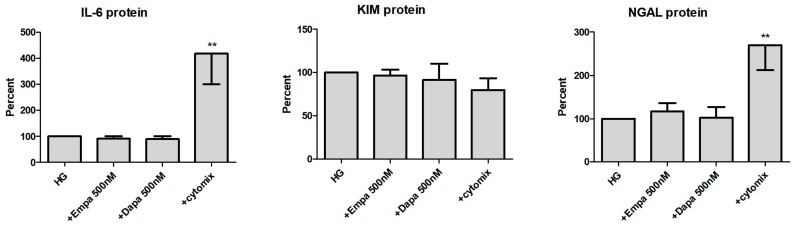
Effect of empagliflozin and dapagliflozin on the release of pro-inflammatory and injury proteins. Cells were grown in a 24-well plate until confluence, serum depleted for 2 h and stimulated in HG for 48 h. Supernatants were then harvested and quantified using commercially available immunoassays (optical density 450/620 nm, calculated as percent in relation to the control (HG), mean ± SD). A mixture of pro-inflammatory cytokines (cytomix) was used as a positive control. (** *p* < 0.01 versus HG, IL-6 *n* = 6, KIM-1 and NGAL *n* = 4).

**Table 1 ijms-21-00391-t001:** Primer used for PCR analyses

Gene	Primer Forward	Primer Reverse	Product Length (bp)	NCBI Reference Sequence
SGLT-2	TGG GCT GGA ACA TCT ATG CC	GTG GAA GGC GTA ACC CAT GA	155	NM_003041.3
IL-6	AAA GAT GGC TGA AAA AGA TGG ATG C	ACA GCT CTG GCT TGT TCC TCA CTA C	150	NM_000600.4
KIM-1	CAG TGG CGT ATA TTG TTG CCG	CAG TCG TGA CGG TTG GAA CA	134	NM_001173393.2
NGAL	GAC CCG CAA AAG ATG TAT GCC	CTC ACC ACT CGG ACG AGG TA	197	NM_005564.4
β-Actin	ACT GGA ACG GTG AAG GGT GAC	AGA GAA GTG GGG TGG CTT TT	169	NM_001101

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
