# Peer review of "No Cytotoxic and Inflammatory Effects of Empagliflozin and Dapagliflozin on Primary Renal Proximal Tubular Epithelial Cells under Diabetic Conditions In Vitro"

_ijms, 2020, doi:10.3390/ijms21020391_

Round 1
Reviewer 1 Report
I have no comments. Wonderful study! Congratulations!
Author Response
ANSWER: We thank the reviewer for the friendly evaluation of our work.
Reviewer 2 Report
I had the opportunity to review the manuscript entitled “No Cytotoxic and Inflammatory Effects of Empagliflozin and Dapagliflozin on Primary Renal Proximal Tubular Epithelial Cells under Diabetic Conditions in Vitro” by Patrick C. Baer, et al who are trying to publish it in International Journal of Molecular Science.
No concerns with the rationale, the methods and the presentation, they are ok.
Main concern is that the study does not completely responds to the posed questions. This is, the authors incubate the PTC cells with three different concentrations of both gliflozins, the highest dose considered as the plasma therapeutically concentration. They assume that the toxicity comes from the contact of the drug with PTC cells. Also, ramipril and HCT were just in contact with PTCs. Some effects could happen, but as authors discus in the Introduction and Discussion, there are several pathophysiologic mechanisms proposed in the literature to understand the toxicity, as for instance high osmolality, elevated uric acid concentration, elevated fructose concentration…. PTCs should be stressed incubating with the drugs in hyperosmotic media, high concentrations of uric acid or fructose, and supratherapeutic concentrations of the drug compounds, in the presence or not of high glucose media. The supratherapeutic concentrations of the drug should mimic the accumulation of the drug in conditions of renal failure.
Author Response
ANSWER: We absolutely agree with the Reviewer. We are currently working on the influence of high fructose and high osmolarity compared to high glucose in the cell culture medium. Cyrrently, we only have initial results on the cytotoxicity of the various sugars and no results on the influence of gliflozins under these conditions (in addition to high Fructose or high osmolarity). First results show an increased cytotoxicity after culture in medium with high fructose (figure below). We look forward to publish these results within the next 12-18 month in an additional manuscript. In these conditions, we also investigate supratherapeutic concentrations of the gliflozins (e.g. 2 µM). Therefore, we also added a sentence at the end of the discussion session.
We added an additional figure showing cytotoxicity in vitro (XTT assay) as a pdf (only for the answer, not included in the revised manuscript).

Reviewer 3 Report
This is an interesting in vitro investigation of the cytotoxic and inflammatory effects of two inhibitors of SGLUT-2, empagliflozin and dapagliflozin. The cell model is well designed for the choice of human renal proximal tubular epithelial cells tested under diabetic conditions using a medium with high glucose content (450 mg/dL). The assays conducted to evaluate cell viability, proliferation, cytotoxicity, expression of kidney injury markers NGAL and KIM-1, inflammation (IL-6) and oxidative stress revealed that the two drugs did not trigger cell injury in vitro.
Some points might be better elucidated according to the following suggestions.
It is known that gliflozins have a good overall tolerability (Di Lullo L et al, Diabetes Metab Syndr. 2017, doi: 10.1016/j.dsx.2017.03.005), although some adverse events have been described, namely AKI, acute tubular necrosis, genital infections and bone fractures. In the introduction section, the reported risk of bone fractures associated with glifozin intake (reference 3) is still unclear, since the increased incidence of bone fractures, firstly reported with dapaglifozin and canaglifozin, was not confirmed in the following RCT (Perkovic V et al, N Engl J Med. 2019, doi: 10.1056/NEJMoa1811744; Wiviott SD et al, N Engl J Med. 2019, doi: 10.1056/NEJMoa181238). Indeed, the limited study follow-up (ranging from 6 months to 2 years) preclude a definitive answer on the impact of these compounds on mineral metabolism changes, especially in high-risk subgroups of patients, asthose with CKD in 3-5 stage (Cianciolo G et al, JBMR Plus. 2019 doi: 10.1002/jbm4.10242) At page 3, the authors mention a drug safety communication by FDA of June 2016 on the higher risk of AKI associated with gliflozin therapy (reference 9), based on total 101 reported cases between March 2013 and October 2015. In this alert, patients with diabetes and heart failure, chronic renal insufficiency, and/or decreased circulating blood volume were identified as risk group, together with those under ACE inhibitors, diuretics and/or nonsteroidal anti-inflammatory therapy. Later, in November 2019, also the German Drug Commission established discontinuation of gliflozins treatment in hospitalized patients after major surgery or acute serious illness. However, despite these relevant safety issues, this point is still under investigation and no firm conclusion can be currently drawn. A recent post-hoc analysis from the EMPA-REG OUTCOME trial (Mayer GJ et al, Kidney Int. 2019 doi: 10.1016/j.kint.2019) revealed the risk of AKI may be actually lower with SGLT2 inhibitors. Indeed, in a large international, real-world study of patients with type 2 diabetes, initiation of SGLT2 inhibitor therapy was associated with a slower rate of kidney function decline and lower risk of major kidney events compared with initiation of other glucose-lowering drugs. (Heerspink HJL et al, Lancet Diabetes Endocrinol2020, https://doi.org/10.1016/S2213-8587(19)30384-5 8: 27–35). In this work, the authors conclude that the two tested antiglycemic agents at therapeutic concentrations have no direct toxic effect on primary renal proximal tubular cells exposed to a diabetic mileu, and might also exert a renoprotective effect through oxidative stress lowering, an effect that appear to be more pronounced with dapagliflozin. These results confirm that glifozins are a varied class of drugs, with effects related to the single agent rather than a class effect. Moreover, in my opinion, these data a should be also commented in view of the fact that diabetes itself is a systemic pro-inflammatory condition, involving not only kidney, but also cardiovascular system and bone, or more precisely the kidney-heart-bone axis (Tousoulis D et al, Curr Med Chem 2015, doi: 10.2174/0929867322666150415145814; Mazzaferro S et al, Nephrol Dial Transplant. 2018, doi: 10.1093/ndt/gfy115).In summary, this is an interesting, well-written and well-designed work and I have no concerns regarding the presentation of methods and results. It that might benefit from some clarifications in the introduction and discussion sections in light of the abovementioned points and the suggested recent references to be added.
Author Response
ANSWER: We are grateful for these valuable advices, and we thank the Reviewer for helping in giving a more accurate introduction and discussion of our results.
As requested by the reviewer, we now added additional sentences to clarify the points raised by the reviewer. We added these sentences mainly in the Introduction section, but also in the discussion.
We also added all new references proposed by the reviewer to the manuscript.
Round 2
Reviewer 2 Report
I understand and accept the honesty of the authors in their answer.
The manuscript and the study itself is absolutely correct, well described and performed, but it only shows the contact of the drug with the PTCs where nothing happens. As I said in my first review, it was foreseeable. If not, gliflozines should be toxic drugs themselves, and they wouldn't have made it to the clinics.
Toxicity of the drug happens occasionally in some scenarios. So, studies have to be done in this scenarios, which answer questions to the physicians.
Author Response
We absolutely agree with the reviewer that this problem should be investigated. Therefore, we added new sentences at the end of the discussion section on this topic and hope this will answer the comment in an appropriate manner. Whereas we could not describe a cytotoxic effect of the gliflozins in the settings used, the current results are one more step to decipher the effects of gliflozins on epithelial cells and are, therefore, important for the scientific community, precisely because of the described induction of acute kidney injury.
We added the following paragraph (page 11):
“Nevertheless, further future studies are needed to investigate potential specific factors leading to tubular injury. First, the effects of incubation in high concentrations of fructose and in hyperosmotic media, both in the presence of gliflozins and high glucose media, should be investigated. High fructose, in vivo generated due to consumption of high fructose dietary products, may also affect tubular epithelial cells [38]. Dietary fructose has been shown to cause tubular injury in vitro and in vivo [21,39]. In addition, an increment in osmotic pressure, in vivo generated by dehydration, may cause activation of the polyol pathway also leading in the accumulation of fructose [16,20]. On the other hand, high concentrations of uric acid could be another factor involved [18], whereas it has been shown that canagliflozin therapy decreased serum uric acid in patients with type 2 diabetes [40]. Finally, the in vitro effects of supratherapeutic gliflozin concentrations that may mimic the accumulation of the drug in conditions of renal failure in vivo should also be investigated in future studies.“
Furthermore, we added 3 new references regarding the effects of high fructose and uric acid.
As noted in the previous answer we are currently working on the influence of high fructose and high osmolarity in the cell culture medium. We look forward to publishing these results within the next 12-18 months. Currently, we only have initial results on the cytotoxicity of the various sugars and no results on the influence of gliflozins under these conditions (in addition to high Fructose or high osmolarity). Our first results showed increased cytotoxicity after culture in medium with high fructose.

Round 3
Reviewer 2 Report
No more suggestions